# Optimizing Business-to-Business Customer Satisfaction Analysis through Advanced Two-Stage Clustering: Insights from Industrial Parks

Jian Wang and Lingling Yue *

School of Management, Shanghai University, Shanghai 200444, China; jwang@t.shu.edu.cn
* Correspondence: 17860733660@shu.edu.cn; Tel.: +86-17860733660

**Abstract:** Traditional research on customer satisfaction (CS) estimation has focused on the business-to-customer (BTC) business mode. Customers in the BTC mode have been assumed to be familiar with the full range of services or products and to be able to make estimations of their CS. However, in the business-to-business (BTB) mode, diverse services have often been required and provided. It may be difficult to find members who have experience with all kinds of services or to generate common CS estimation results supported by different members. In this study, the difference between BTC and BTB was verified using structural equation modeling (SEM), and a model of CS estimation was developed with respect to BTB. The empirical results show that perceived service quality has no direct impact on enterprise satisfaction, indicating that traditional models are limited. A two-stage clustering algorithm was adopted to optimize the traditional CS evaluation model based on SEM, i.e., (1) K-nearest neighbor (KNN) classification and (2) density-based spatial clustering of applications with noise (DBSCAN). In order to verify the feasibility of the proposed model, CS with respect to six industrial parks was estimated empirically. The results show that the proposed model can improve the results of CS estimation compared with the results obtained using traditional methods. During the clustering process, each park generated and eliminated a certain number of noise points to optimize the satisfaction evaluation results. Specifically, park A generated and eliminated seven noise points, while park C generated and eliminated five noise points. The results of the satisfaction evaluation of each park obtained using the proposed model are more realistic, i.e., park A > park B > park C > park E > park D > park F. The proposed model extends the existing research on CS estimation in theory and can support applications in the BTB business mode.

**Keywords:** customer satisfaction evaluation; BTB; two-stage clustering; industrial parks

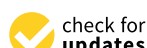



## 1. Introduction

With the emergence of individualized customer needs, an enterprise or organization's service level has become crucial for its sustainable development. Determining how to reasonably evaluate the service level of an enterprise or organization is another challenge. Customer satisfaction evaluation (CSE) can be found in a wide range of industries, e.g., manufacturing, logistics, tourism, hospitality, etc. Questionnaire surveys and structural equation modeling (SEM) have often been adopted in CSE. There is a general assumption in the surveys that the customer completing the questionnaire can give a reasonable perceptual evaluation because they are familiar with the product or service provided. This is possible in the business-to-customer (BTC) mode, because the product or service is provided to the customers directly. Unlike in the BTC business mode, customers in the business-to-business (BTB) mode are enterprises or organizations with complex needs, including diverse services. In recent years, the needs of customers in the BTB mode have become increasingly complex. Providers must continuously improve their services to meet the needs of enterprise users in multiple dimensions [1]. However, it may be difficult for a single employee in a customer enterprise to experience all of the services and, therefore, it is difficult to give an accurate

evaluation [2]. Although group discussion can be used for the evaluation, it may be costly and difficult to generate the common results shared by groups.

The challenge can be found in different BTB cases, i.e., industrial parks, electronic commerce platforms, etc. For example, the emergence of an industrial park can address the infrastructural, managerial, environmental, social, and economic aspects of a region in order to make it more sustainable. Industrial parks provide diverse services to the enterprises that buy or rent the buildings or other infrastructure of the parks. Customer enterprises may be called settled enterprises of parks, and can choose whether or not to settle in based on their preliminary research and expectations of the parks. This is the core component of CSE, and directly affects the core competitiveness of industrial parks. Based on the above research background, the research questions of this study are as follows: (1) Does the traditional BTC CSE model apply to the BTB mode in the context of diverse services? (2) What methods are used to optimize the results of individual user satisfaction scoring in the BTB mode, considering the difference between group scoring and individual scoring?

The aim of this study was to address the above problems by developing a new model and introducing a novel algorithm to propose a CSE model that is applicable to the BTB mode. The remaining sections are organized as follows: Section 2 summarizes the existing research on CSE and identifies the research gaps for BTB. Section 3 proposes a BTB CSE model considering diverse services. This model combines SEM and a clustering algorithm. Section 4 verifies the practical feasibility of the BTB CSE model through the case of industrial parks. Conclusions and future work are summarized in the Section 5.

## 2. Literature Review

### 2.1. Definition and Measurement of CS

Customers are usually defined as recipients of products and services. In ISO 9000 [3], customer satisfaction (CS) is defined as the perception of the extent to which their requirements have been met. From a service perspective, CS refers to the overall level of satisfaction with the tangible and intangible services provided by an organization. From the perspective of the buyer–seller relationship, CS is an evaluation of whether the product meets or exceeds the expectations of the customer when the product is purchased or used [4]. With the transformation of the economy from a seller's market to a buyer's market, CS has become a prerequisite for the survival and development of enterprises. Higher CS can increase the return on investment and repurchase, while lower CS has a negative impact on the organization's competitiveness [5].

Several famous models have been established for CSE, e.g., the Swedish Customer Satisfaction Barometer (SCSB) developed by Statistics Sweden in 1989, the American Customer Satisfaction Index (ACSI) model, the Japanese Customer Satisfaction Index (JCSI) model, the European Customer Satisfaction Index (ECSI) model, the China Customer Satisfaction Index (CCSI), etc. SEM has often been adopted in the above models. It has been widely applied in social sciences research with some variables that cannot be measured directly, e.g., social status, satisfaction, loyalty, etc.

### 2.2. Traditional BTC Customer Satisfaction Evaluation

Most traditional studies on CSE have focused on the BTC mode. Three methods are commonly used for CSE, i.e., SEM, SERVQUAL, and service importance estimation.

The first method involves using SEM to analyze the factors affecting CS. SEM can analyze the relationships between multiple independent and dependent variables at the same time, i.e., the antecedent variables and the outcome variables.

The antecedent variables of CS include customer expectations (CEs), perceived quality (PQ), and perceived value (PV). From the pathway relationship of PQ to PV, Tong et al. [6] explored satisfaction with community health education among residents in China. The study showed that PQ had the strongest correlation with PV. Zhou et al. [7] applied the ACSI model for the first time to investigate the primary factors affecting the travel services of passengers in China's online ride-hailing industry. The study showed that the PQ and

PV of online taxis had a significant positive impact on tourist satisfaction. The variable of corporate image also plays a significant role in SEM. Dash et al. [8] investigated the impact of brand identity and brand image on CS. Liu et al. [9] expanded the ACSI model by considering corporate image as an important variable of CS. Unlike in previous studies, Xie et al. [10] used service image as an outcome variable and proposed a new perspective that public satisfaction has a positive impact on service image.

The outcome variables of CS include customer complaints (CCs), customer loyalty (CL), etc. For example, Yilmaz and Ari [11] explored the factors influencing the quality of high-speed rail services based on SEM. The study showed that improving CS leads to fewer CCs and higher CL. Ahmed et al. [12] examined the economic aspects of business operations in the airline industry and concluded that an immediate response to CCs enhances CS and CL. The above research findings confirm that traditional CSE models are generally applicable in the BTC mode. These studies mainly used traditional CSE models to distribute questionnaires to the end customer and analyze the influencing factors. BTC CSE studies based on SEM have been widely used in various fields.

The second method is based on five attributes of SERVQUAL, i.e., tangibility, responsiveness, assurance, reliability, and empathy. For example, Miranda et al. [13] used an extended model of SERVQUAL to analyze the impact of different combinations of service quality dimensions on CS. In recent years, scholars have proposed a method that combines SERVQUAL service attributes with SEM to optimize CSE. For example, Wattoo and Iqbal [14] merged the SERVQUAL and ACSI models to propose that service quality improvements can boost consumer satisfaction with e-commerce platforms. This method also focuses on CS studies in the BTC mode.

The third method involves conducting CSE based on the importance of service content. As far back as twenty years ago, Gustafsson and Johnson [15] suggested that the degree of importance customers attach to products and services has an impact on satisfaction and loyalty. The common methods of service importance estimation include the expert evaluation method, Kano model, relative weighting analysis (RWA), etc. In addition, Hsieh [16] proposed Beyond Multiplication, which is different from multiplication scores. This method was conducted through face-to-face interviews with each respondent to obtain their satisfaction and importance scores for each service. However, some scholars have argued that the assignment of importance in assessing CS is subjective. For example, Campbell et al. [17] found no empirical evidence supporting the idea that importance has a significant impact on measuring satisfaction with quality of life. In the field of application, the service importance estimation method is mainly applicable to the end-customer-oriented BTC mode. Considering the high cost of group discussion, the method is applied less in the BTB mode.

The above three CSE methods are mostly oriented to the end customer, based on extended models of SCSB, ACSI, ECSI, as well as the SERVQUAL and Kano models. These methods are mainly used to analyze the influencing factors of CS and the correlations among the variables.

*2.3. BTB Customer Satisfaction Evaluation*

In recent years, the market size of China's industrial parks and e-commerce platforms has steadily grown due to the development of science, technology, and the internet. According to the National Bureau of Statistics (NBS), the market size of China's e-commerce platforms in 2022 grew by 3.5% from the previous year. In this article, e-commerce platforms and industrial parks are used as case studies to examine BTB CSE. However, previous studies have not developed a theoretical model of CSE for the BTB mode. Scholars still refer to the traditional BTC CSE model. Previous studies did not consider whether the traditional CSE models applicable to end customers are still applicable to customer enterprises. Table 1 provides a review of BTB CSE studies. The first column indicates the areas of research. The second column indicates the survey respondent, which can reflect the familiarity with diverse services. The third column indicates the level of service diversification in these

studies. "High" represents a tendency towards diverse services, and "Low" represents a tendency towards homogenized services. The fourth column indicates the research methods. The final column indicates the literature source.

**Table 1.** A review of BTB customer satisfaction research.

| Research Area | Respondent | Degree of Service Diversity | Research Methods | Literature Sources |
|---|---|---|---|---|
| Financial services virtual community | Platform subscribers | Low | Structural equation modeling | [18] |
| BTB cross-border e-commerce platform | Industrial experts | High | Modified Delphi technique and Kano model | [19] |
| Private science and technology park | Managers and employees of enterprises in the park | High | FAHP and Fuzzy-DEMATEL | [20] |
| Science and technology park | Managers of settled enterprises | High | Semi-structured method interview | [21] |
| BTB social media platform | Employees of the buyer's enterprise | Low | Partial least squares structural equation modeling | [22] |
| Automobile industrial park | Head of enterprise | Medium | Standard impact loss method and Kano model | [23] |
| International industrial park | UNIDO | High | Eco-industrial parks assessment tool | [24] |

### 2.3.1. BTB E-Commerce Platform Customer Satisfaction Evaluation

In the BTB mode, the third-party e-commerce platform can provide diverse services, e.g., publishing supply and demand information, facilitating customer and supplier transactions, etc. It is necessary to measure the satisfaction with third-party platforms, which can promote the long-term development of both enterprises and platforms. For example, Chompis et al. [18] used SEM to assess enterprise users' satisfaction with a financial service platform. Ho and Chuang [19] explored the service quality of BTB cross-border e-commerce platforms using a modified Delphi method and Kano model. The study identified and prioritized the key service quality attributes of BTB cross-border e-commerce platforms. Previous studies have commonly relied on the evaluation results of managers and relevant experts from enterprises to represent the evaluation results of the entire organization. This approach may introduce bias and subjectivity.

### 2.3.2. Industrial Park Enterprise User Satisfaction Evaluation

Industrial parks and settled enterprises are a common BTB mode. The satisfaction of settled enterprises is influenced by the quality of the service provided in industrial parks. In this section, industrial parks are used as a case study to examine the satisfaction levels of customer enterprises for diverse services. For example, Weng et al. [20] used FAHP and Fuzzy-DEMATEL methods to survey the managers and employees of the enterprise in a park. The aim was to analyze the critical success factors for establishing a private science and technology park. Lecluyse and Knockaert [21] used semi-structured interviews to explore the satisfaction of settled enterprises in a science and technology park. The study showed that achieving CS is contingent on settled enterprises' expectations, perceived service quality, and pre-settlement achievements. The personalized demands of settled enterprises have motivated the increasing diversity of park services. However, in previous studies, the selection of respondents involved individual users instead of enterprise users.

### 2.4. Comparison of BTC and BTB Customer Satisfaction Evaluation

The research and application of satisfaction in the BTC mode have matured. Due to the differences between BTB and BTC, traditional CSE models are not completely applicable to the BTB mode. The differences between the two modes are as follows. (1) Service diversity. According to service contact theory [25], CS is affected by the entire process by which customers encounter diverse services. In the BTC mode, the end customer is more familiar with diverse services, so it is more reasonable to give satisfaction evaluation responses. In the BTB mode, the satisfaction of one employee cannot substitute for the satisfaction of the entire organization. (2) Demand dynamics. According to life-cycle theory, unlike the end customer in the BTC mode, the needs of customer enterprises at different development stages in the BTB mode are dynamic. For example, Ferreira et al. [26] showed that enterprises at different developmental stages have different physical, labor, and financial levels. These enterprises also have different needs for diverse services. Identifying the needs of customer enterprises for diverse services at different development stages can increase their repurchase intention. In this article, we focus on comparing the differences in CSE between BTC and BTB modes in terms of service diversity.

The emergence of personalized and differentiated customer demands has led to the diversification of the services provided by enterprises or organizations. The level of service contact throughout the process determines the customer's satisfaction with the service. The entire process of receiving services involves interactions with service personnel and interactions with equipment and facilities [25]. The customer's actual contact in the service process forms a perception, and the perceived level of service is used to assess the service quality of the enterprise or organization. In the BTC mode, the end customer is more familiar with the products or services provided by the enterprise or organization, so the satisfaction evaluation results given are more reasonable. Previous BTC CSE studies have involved further extended research based on traditional evaluation models such as SCSB, ACSI, ECSI, SERVQUAL, and others. In the BTB mode, the recipient of the product or service is another enterprise with complex needs. Due to their different interactions with diverse services, individual employees in the enterprise may generate different satisfaction feelings. Therefore, it is not appropriate to substitute the service level perceived by individual employees for the service level perceived by the enterprise as a whole. For example, in the study of CSE in industrial parks, different employees of enterprises have different levels of exposure to the diverse services provided by the park, such as supporting facilities, talent services, science, and innovation. However, in questionnaire analysis, individual employee satisfaction evaluations are typically used instead of evaluations for the entire enterprise. The satisfaction evaluation results obtained in this way are biased.

This article further explores the applicability of traditional CSE models in the BTB mode by considering the characteristics of service diversity.

### 3. Optimizing BTB Customer Satisfaction Evaluation with a Clustering Algorithm Considering Service Diversity

#### 3.1. Purpose of the Two-Stage Clustering Algorithm

Econometric measures, the fuzzy comprehensive evaluation method, principal component analysis, and the gray system method have commonly been used to evaluate CS in previous studies. These methods were used to address many issues in previous CSE research [27–31]. However, services are more diverse in the BTB mode. During the data collection phase, the representativeness of respondents was not taken into account when addressing the issue of respondent representation in the BTB mode, i.e., "It is not reasonable to replace enterprise user satisfaction evaluation results with individual user satisfaction evaluation results". New evaluation methods should be considered.

Clustering algorithms have the advantages of "reducing complexity" and "increasing consensus" when processing data, and they have been widely used in the fields of internet search, face recognition, and information security. Our research can be used to apply a

clustering algorithm to optimize satisfaction evaluation results, which will improve data credibility and consensus. This will have significant application and promotional value. Commonly used clustering algorithms include the K-means clustering algorithm, the K-nearest neighbor classification algorithm, and the DBSCAN clustering algorithm. However, the K-means algorithm has some drawbacks, such as the requirement to predetermine the number of clusters and the possibility of inaccurate clustering results.

In order to incorporate the advantages of clustering algorithms in improving the consensus of satisfaction evaluation results, a two-stage clustering algorithm combining the KNN classification algorithm and DBSCAN clustering algorithm was used in this study to optimize the BTB CSE. The algorithm is based on the following two points.

1.  Density-based spatial clustering of applications with noise (DBSCAN) is a classical density-based clustering algorithm. It has the advantage that the number of clusters does not need to be predetermined and can effectively identify arbitrarily shaped datasets and outliers [32]. The key to using this algorithm is determining the input parameters, i.e., radius and threshold. The clustering algorithm can be used to identify and eliminate the noise point data from individual user satisfaction evaluation results. The final clustering results obtained through this method exhibit a higher consensus.
2.  The K-nearest neighbor (KNN) algorithm is commonly used to determine the classification of a sample by calculating its distance to all of the samples. In addition, the algorithm reduces the iterative experimental process of input parameters in the DBSCAN clustering algorithm by confirming the input radius. The human interference in this process is reduced. This is mainly achieved by using the K-dist method and calculating the Euclidean distance. The advantage of this algorithm is that it can be used to optimize the DBSCAN algorithm.

### 3.2. Evaluation Process of Customer Satisfaction Evaluation Optimization Model for BTB

To address the limitations of traditional CSE models based on SEM in measuring satisfaction with diverse services, an evaluation method combining SEM and a two-stage clustering algorithm was applied in this study. The evaluation process of the optimization model for BTB CSE is shown in Figure 1, which includes the following important steps:

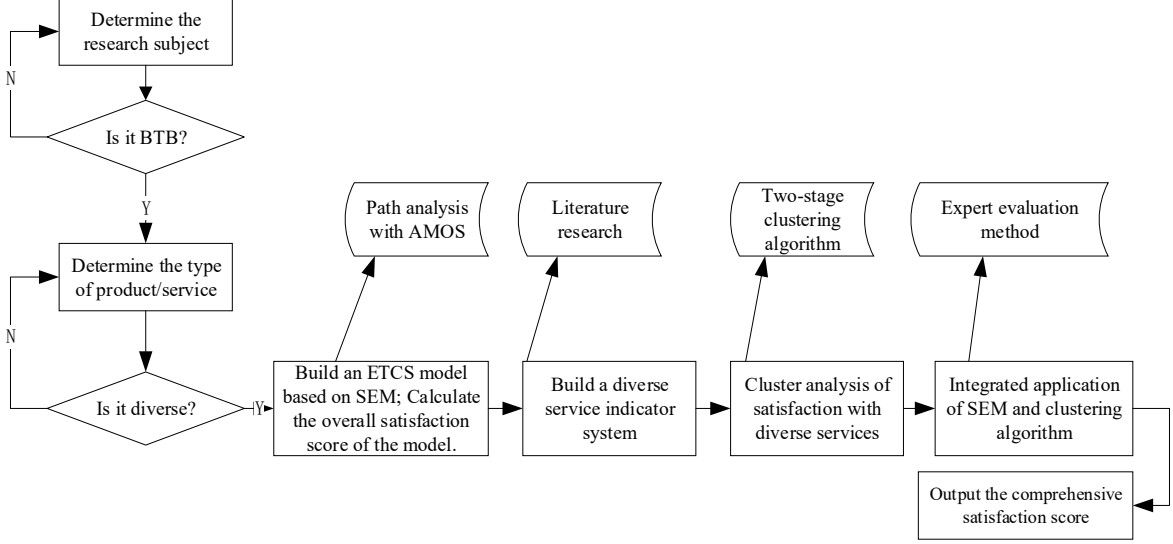

**Figure 1.** Flowchart of customer satisfaction evaluation optimization model for BTB with diverse services.

1.  Analysis of service diversity

It is determined whether the research subject belongs to the BTB business mode, i.e., an enterprise or organization that provides a product or service on one end and an enterprise

or organization that uses the product or service on the other end. The diversity of products or services offered is analyzed.

2.     Hypothesis testing for SEM

In order to verify the limitations of the traditional models in the BTB mode that provides diverse services, an SEM is constructed using the traditional CSE models to examine the relationships among the variables. Using the standardized path coefficients between the latent variables and the measurement items, the satisfaction scores of the traditional CSE model based on SEM are calculated.

In this study, a customer satisfaction index model for enterprise users (ETCS) was constructed, as shown in Figure 2. In contrast to CS, which is oriented towards the end customer, enterprise satisfaction refers to the level of satisfaction of the entire customer enterprise. The measurements in the model include 12 items: (1) overall service level and perceived personalized service demand as measurement items for perceived service quality, (2) overall service expectation, personalized service demand expectation, and guaranteed service demand expectation as measurement items for enterprise service expectation, (3) price relative to quality, quality relative to price, and sense of accomplishment as measurement items for perceived service value, (4) overall satisfaction and developmental satisfaction as measurement items for enterprise satisfaction, and (5) expanding willingness to utilize and willingness to recommend as measurement items for enterprise loyalty.

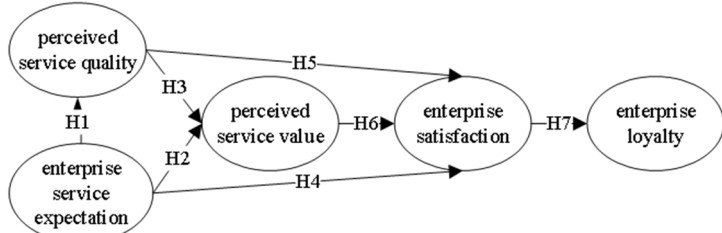

**Figure 2.** ETCS model framework.

The paths in the model are explained as follows: H1: Enterprise service expectation has a positive direct impact on perceived service quality. H2: Enterprise service expectation has a positive direct impact on perceived service value. H3: Perceived service quality has a positive direct impact on perceived service value. H4: Enterprise service expectation has a positive direct impact on enterprise satisfaction. H5: Perceived service quality has a positive direct impact on enterprise satisfaction. H6: Perceived service value has a positive direct impact on enterprise satisfaction. H7: Enterprise satisfaction has a positive direct impact on enterprise loyalty.

3.     Cluster analysis considering service diversity

Since service diversity is not considered in the second step, in the third step, a diverse service indicator system is constructed. In order to solve the problem of "it is not reasonable for individual users to evaluate the satisfaction of diverse services", a two-stage clustering algorithm based on the combination of KNN and DBSCAN is used to optimize the satisfaction results. By identifying and eliminating outliers, the clustered satisfaction results have a higher consensus.

The steps used to achieve this are as follows. (1) Building a diverse service indicator system. To reflect the characteristics of service diversity in BTB mode, a diverse service indicator system is constructed. (2) Factor analysis for dimensionality reduction. A two-stage clustering algorithm is best suited for clustering low-dimensional data. Therefore, factor analysis is used to downscale the diverse services before density clustering. (3) Two-stage clustering optimizes the satisfaction evaluation results. In the first stage, distances are computed using the KNN algorithm to find the inflection point as the input radius $\varepsilon$. The value of K is confirmed by referencing [33], K = 4. In the second stage, the DBSCAN clustering

algorithm is utilized to identify and eliminate the noise points to reach a satisfaction consensus. This step first requires the determination of the threshold MinPts, which is selected as MinPts = 4 based on a priori knowledge [32]. Subsequently, the dimensionality-reduced dataset D, radius $\varepsilon$, and threshold MinPts are imported into Python 3.9 as input parameters for clustering to identify and eliminate noise points. Finally, the K-means algorithm is employed to determine the center point of each cluster, and the outcome is presented in a 3D stereogram.

4.     Integrated application of SEM and clustering algorithm

In this step, the expert evaluation method is used to assign weights to the two methods, i.e., SEM and the two-stage clustering algorithm, and calculate the comprehensive satisfaction score.

In this study, a CSE optimization model for diverse services was developed, which can be applied to enterprise users.

## 4. A Case Study of CSE for BTB

Industrial parks and settled enterprises are a common BTB mode. As case study objects, we chose six industrial parks located in one of China's more developed cities. These parks ranked among the top 50 in terms of characteristics and the top 20 in terms of comprehensive strength. The criteria for selecting industrial parks in this article were as follows: (1) number of settled enterprises in each park; (2) size of industrial parks; (3) date of establishment of industrial parks. For example, parks A, B, C, and F all have more than 2000 settled enterprises. Parks D and E are larger, to cater to many types of settled enterprises. All the parks have been established for more than 10 years, including parks A, B, C, D, E, and F.

### 4.1. Data Collection

The survey was conducted through a questionnaire. Park managers and experts developed a CS questionnaire for industrial parks. There were two types of questionnaires. Questionnaire 1 was conducted based on the SEM and Questionnaire 2 was conducted for diverse service components. The questionnaires were distributed through the online platform Questionnaire Star to the settled enterprises in each park between the end of October and the end of November 2022. The relevant person in each enterprise completed both questionnaires. The respondents selected for this study included junior employees, junior managers, middle managers, and senior managers of the enterprises.

4.1.1. Questionnaire Design and Collection Based on SEM

Questionnaire 1 was developed with reference to the ETCS model in Figure 2, and the latent variables were measured using a 10-point Likert scale, with "1" indicating very dissatisfied and "10" indicating very satisfied. The measurement items are shown in Table 2. The number of measured variables corresponding to the latent variables included two or three.

**Table 2.** Variable measurement items.

| Latent Variables | Measured Variables | Measurement Items |
|---|---|---|
| Perceived service quality (PQ) | Overall service level | PQ1 |
| | Perceived personalized service demand | PQ2 |
| Enterprise service expectation (SE) | Overall service expectation | SE1 |
| | Personalized service demand expectation | SE2 |
| | Guaranteed service demand expectation | SE3 |
| Perceived service value (PV) | Price relative to quality | PV1 |
| | Quality relative to price | PV2 |
| | Sense of accomplishment | PV3 PV4 |

| Latent Variables | Measured Variables | Measurement Items |
|---|---|---|
| Enterprise satisfaction (ES) | Overall satisfaction | ES1 |
| | Developmental satisfaction | ES2 |
| Enterprise loyalty (EL) | Expanded willingness to utilize | EL1 |
| | Willingness to recommend | EL2 |

After analyzing the recovered questionnaires, those with a higher number of missing responses and repeated options were removed. Ultimately, 255 valid questionnaires were confirmed, resulting in an effective recovery rate of 51.00%.

4.1.2. Questionnaire Design and Collection Based on Diverse Service Components

Park services are diverse, e.g., infrastructure, investment and financing, talent support, etc. In this study, the indicator framework used to assess international eco-industrial parks and the classifications of park services by different scholars were referenced [20,24]. Three aspects of the KANO model were also considered, i.e., basic, performance, and motivational factors. A diverse service indicator system including 10 services of the park was constructed (refer to Table 3).

**Table 3.** Classification of diverse service indicators in parks.

| Number | Indicators | Serial Number | Measurement Items |
|---|---|---|---|
| 1 | Infrastructure services | IS | IS1; IS2; IS3 |
| 2 | Basic property | BP | BP1; BP2; BP3 |
| 3 | Business services | BS | BS1; BS2; BS3 |
| 4 | Security management | SM | SM1; SM2; SM3 |
| 5 | Institutional safeguards | SG | SG1; SG2; SG3 |
| 6 | Financing services | FS | FS1; FS2; FS3 |
| 7 | Social influence | SI | SI1; SI2; SI3 |
| 8 | Talent development | TD | TD1; TD2; TD3 |
| 9 | Technology innovation | TI | TI1; TI2; TI3 |
| 10 | Specialty services | SS | SS1; SS2; SS3 |

Questionnaire 2 was developed based on the basic characteristics of service quality, including tangibility, responsiveness, assurance, reliability, and empathy. Each service was evaluated using three measurement items, for a total of 30 survey questions. The survey employed a 10-point Likert scale, with "1" indicating very dissatisfied and "10" indicating very satisfied. A total of 500 questionnaires were distributed and completed by the employees of the enterprises. Similarly, after analyzing the recovered questionnaires, those with a higher number of missing responses and repeated options were removed. A total of 279 valid questionnaires were confirmed, resulting in an effective recovery rate of 55.80%.

4.1.3. Basic Information on Interviewed Enterprises

Table 4 displays the basic information about the interviewed enterprises. Approximately 60% of the enterprises have been settled in these parks for over three years, and the settled enterprises belong to a variety of industries. This makes the questionnaire representative.

**Table 4.** Descriptive statistical information from the questionnaire.

| Characterization | Options | Frequency | Percentage |
|---|---|---|---|
| Number of years the enterprise has been in the park | Less than 3 years | 103 | 40.08% |
| | 3–5 years | 48 | 18.67% |
| | 6–10 years | 70 | 27.24% |
| | More than 10 years | 36 | 14.01% |

**Table 4.** *Cont.*

| Characterization | Options | Frequency | Percentage |
|---|---|---|---|
| | Integrated circuit | 11 | 3.94% |
| | Biomedical | 42 | 15.05% |
| | Artificial intelligence | 10 | 3.58% |
| | Electronic information | 26 | 9.32% |
| | Life health | 12 | 4.30% |
| Industry | Automotive | 14 | 5.02% |
| | High-end equipment | 22 | 7.89% |
| | Advanced material | 8 | 2.87% |
| | Consumer fashion | 9 | 3.23% |
| | Service industries | 56 | 20.07% |
| | (sth. or sb) Else | 69 | 24.73% |
| | Less than 3 years | 148 | 53.05% |
| Number of years respondents have worked | 3–5 years | 58 | 20.79% |
| | 6–10 years | 52 | 18.64% |
| | More than 10 years | 21 | 7.52% |

*4.2. CSE Based on SEM in BTB Mode*

4.2.1. Reliability and Validity Testing of Questionnaire

The results of the confidence analysis are shown in Table 5. The overall Cronbach's alpha coefficient for the satisfaction scale is 0.973. Cronbach's alpha coefficients for each latent variable are 0.899 for perceived service quality, 0.952 for enterprise service expectation, 0.955 for perceived service value, 0.814 for enterprise satisfaction, and 0.922 for enterprise loyalty. The coefficients all exceed 0.7, indicating a high degree of consistency. Table 6 shows that the KMO value is 0.950 and the significance level is less than 0.05, indicating its suitability for factor analysis.

**Table 5.** Results of confidence analysis.

| Latent Variables | Number of Items | Cronbach's Alpha | Overall Cronbach's Alpha |
|---|---|---|---|
| PQ | 2 | 0.899 | |
| SE | 3 | 0.952 | |
| PV | 4 | 0.955 | 0.973 |
| ES | 2 | 0.814 | |
| EL | 2 | 0.922 | |

**Table 6.** Test results.

| KMO and Bartlett's Test | | Results |
|---|---|---|
| KMO Sample Suitability Quantity | | 0.950 |
| | Approximate chi-square | 4294.774 |
| Bartlett's test of sphericity | (Number of) degrees of freedom | 78 |
| | Significance | 0.000 |

A validated factor analysis of the model was conducted. To assess the internal reliability of the measurement model, we utilized AMOS 26 to compute the values for factor loading coefficients, composite reliability (CR), and average variance extracted (AVE) for each variable in the ETCS model. Details are shown in Table 7. The CR values of the five latent variables are all above 0.7 and the AVE values are all above 0.5, indicating strong internal consistency of the model.

**Table 7.** Factor loading coefficients.

| Latent Variables | Survey Questions | Standard Load Factors | CR | AVE |
|---|---|---|---|---|
| Enterprise service expectation | SE1: How much does your company expect the park to provide a comprehensive range of services? | 0.949 | 0.957 | 0.882 |
| | SE2: How much does your company expect the park to provide personalized service? | 0.969 | | |
| | SE3: How much does your company expect to be able to use the services of the park anytime, anywhere? | 0.898 | | |
| Perceived service quality | PQ1: How does your company feel about the overall level of service actually provided by the park? | 0.900 | 0.923 | 0.856 |
| | PQ2: To what extent does the park actually meet the personalized service demand of your company? | 0.950 | | |
| Perceived service value | PV1: Relative to the services provided by the park, your company feels that the cost of occupancy is very reasonable. | 0.904 | 0.962 | 0.863 |
| | PV2: Relative to the cost of occupancy, your company is very satisfied with the services provided by the park. | 0.951 | | |
| | PV3: The services provided by the park are very helpful for your company's business development. | 0.963 | | |
| | PV4: The services provided by the park are very helpful for your company to grow in terms of performance. | 0.896 | | |
| Enterprise satisfaction | ES1: Overall, your company is very satisfied with the services provided by the park. | 0.871 | 0.873 | 0.774 |
| | ES2: The services provided by the park are very helpful for your company to develop a competitive advantage. | 0.889 | | |
| Enterprise loyalty | EL1: If your company needs extra office/production space due to business development, will your company prioritize this park? | 0.913 | 0.930 | 0.870 |
| | EL2: Would you recommend this park to your friends if they are in need of a park? | 0.952 | | |

### 4.2.2. Model Fit Testing

Table 8 displays the results of the fit statistics, all of which are acceptable.

**Table 8.** Structural model fit statistics.

| Fit | $\chi^2/\mathrm{df}$ | GFI | AGFI | RMSEA |
|---|---|---|---|---|
| Actual value | 2.114 | 0.931 | 0.883 | 0.066 |
| Standard value | (1,3) | $\geq 0.90$ | $\geq 0.80$ | $\leq 0.08$ |

### 4.2.3. Path Analysis and Hypothesis Testing

The standardized path coefficients among the variables and the model hypothesis results are shown in Table 9 and Figure 3. According to the impact pathways presented in Table 9, it is evident that enterprise service expectation has a positive impact on perceived service quality, thus confirming hypothesis 1. Enterprise service expectation has a positive impact on perceived service value, thus confirming hypothesis 2. Perceived service quality has a positive impact on perceived service value, thus confirming hypothesis 3. Enterprise service expectation has a positive impact on enterprise satisfaction, thus confirming hypothesis 4. Perceived service value has a positive impact on enterprise satisfaction, thus confirming hypothesis 6. Perceived service quality has no direct impact on enterprise satisfaction, meaning that hypothesis 5 is invalid. The standardized path coefficient of enterprise satisfaction on enterprise loyalty is 0.933, indicating that an increase in enterprise satisfaction causes greater enterprise loyalty, thus confirming hypothesis 7.

**Table 9.** Hypothesis results of the model.

| Impact Path | Standardized Path Coefficients | *p* | *p*-Value Range | Hypotheses Results |
|---|---|---|---|---|
| Enterprise service expectation → Perceived service quality | 0.869 | *** | <0.001 | Support for H1 |
| Enterprise service expectation → Perceived service value | 0.214 | 0.002 | <0.01 | Support for H2 |
| Perceived service quality → Perceived service value | 0.766 | *** | <0.001 | Support for H3 |
| Enterprise service expectation → Enterprise satisfaction | 0.210 | 0.012 | <0.05 | Support for H4 |
| Perceived service quality → Enterprise satisfaction | −0.168 | 0.302 | >0.05 | No support for H5 |
| Perceived service value → Enterprise satisfaction | 0.928 | *** | <0.001 | Support for H6 |
| Enterprise satisfaction → Enterprise loyalty | 0.933 | *** | <0.001 | Support for H7 |

Notes: *** $p < 0.001$.

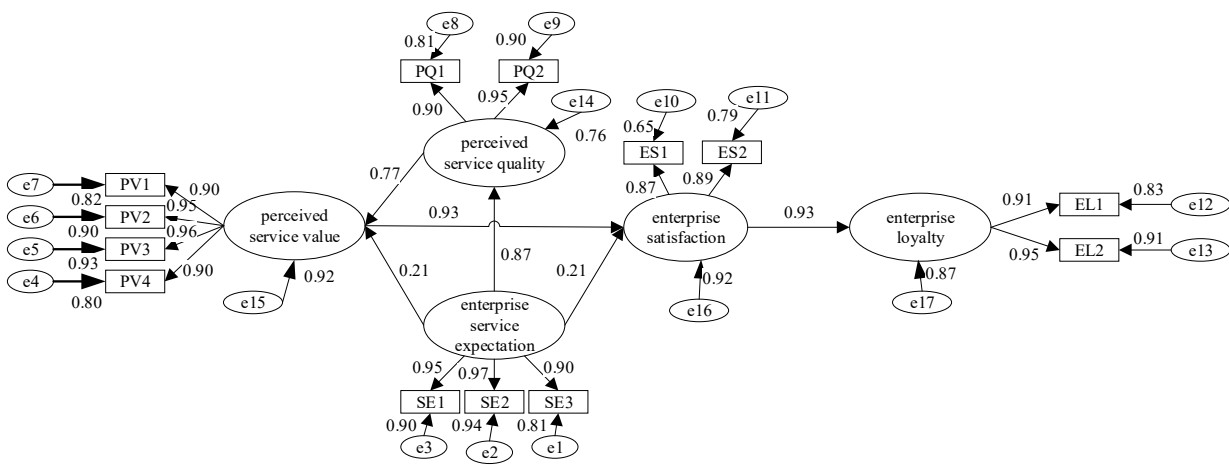

**Figure 3.** ETCS model standardized path analysis results.

According to each standardized path coefficient of the SEM results in Figure 3, we can calculate the weights of each measurement item. The formulas are as follows:

$$\overline{\omega_i} = \frac{\omega_i}{\sum_1^n \omega_i} \tag{1}$$

where $\omega_i$ denotes the standardized path coefficient of the ith measurement item, *n* denotes the number of measurement items in the corresponding latent variable, and $\overline{\omega_i}$ denotes the weight of the ith measurement item in the corresponding latent variable.

$$L_j = \sum_1^n \overline{\omega_i}\, \overline{M_i} \tag{2}$$

where $\overline{\omega_i}$ denotes the weight of the ith measurement item in the corresponding latent variable, $\overline{M_i}$ denotes the mean value of the ith measurement item, and $L_j$ denotes the score of the jth latent variable.

$$ETCS = \frac{\sum_1^m L_j}{m} \tag{3}$$

where *ETCS* denotes the customer satisfaction index score for enterprise users and *m* denotes the number of latent variables. CS results for enterprise users based on the SEM are shown in Table 10.

**Table 10.** CS results for enterprise users based on SEM.

| Latent Variables | Measurement Items | Standardized Path Coefficients | $\overline{\omega_i}$ |
|---|---|---|---|
| Enterprise service expectation | SE1 | 0.949 | 0.337 |
| | SE2 | 0.969 | 0.344 |
| | SE3 | 0.898 | 0.319 |
| Perceived service quality | PQ1 | 0.900 | 0.486 |
| | PQ2 | 0.950 | 0.514 |
| Perceived service value | PV1 | 0.904 | 0.243 |
| | PV2 | 0.951 | 0.256 |
| | PV3 | 0.963 | 0.259 |
| | PV4 | 0.896 | 0.241 |
| Enterprise satisfaction | ES1 | 0.871 | 0.495 |
| | ES2 | 0.889 | 0.505 |
| Enterprise loyalty | EL1 | 0.913 | 0.490 |
| | EL2 | 0.952 | 0.510 |

According to Table 10, we can calculate the CS index score of each park. The score for park A is 9.251, the score for park B is 8.879, the score for park C is 8.909, the score for park D is 8.767, the score for park E is 8.339, and the score for park F is 8.770. The ranking of satisfaction for each park is as follows: park A > park C > park B > park F > park D > park E.

4.2.4. Data Analysis

As a whole, the hypothesized relationships of H1, H2, H3, H4, H6, and H7 are valid. This is the same as previous findings. The result confirms that CSE research based on SEM still has high applicability in the BTB mode. However, the positive impact of perceived service quality on enterprise satisfaction is not significant. This is different from the findings of previous BTC CSE studies. Perceived service quality focuses on how enterprises actually feel about service quality. Combined with the service contact theory [25], the services provided by the park are diverse, and the degree of contact and real feelings of individual enterprise employees towards the diverse services are different. It is not reasonable to replace all of the enterprise satisfaction evaluation results with individual enterprise employees' satisfaction evaluation results. Moreover, the measurement of diverse services is not reflected in the ETCS model. Therefore, while the traditional CSE model is largely applicable to the BTB mode, it must be improved to compensate for the shortcomings of the traditional model. This is the reason why a clustering algorithm was introduced in this study.

*4.3. CSE Based on Two-Stage Clustering in BTB Mode*

4.3.1. Reliability and Validity Testing of Questionnaire

The reliability and validity of the questionnaire were tested. The results are shown in Tables 11 and 12. Cronbach's alpha coefficients for 10 service indicators in the questionnaire all exceed 0.8. The reliability of the questionnaire is high and the KMO value of 0.956 indicates that the factor analysis is reasonable.

**Table 11.** Results of confidence analysis.

| Indicators | Number of Items | Cronbach's Alpha | Overall Cronbach's Alpha |
|---|---|---|---|
| IS | 3 | 0.889 | |
| BP | 3 | 0.902 | |
| BS | 3 | 0.927 | |
| SM | 3 | 0.952 | |
| SG | 3 | 0.897 | 0.972 |
| FS | 3 | 0.876 | |
| SI | 3 | 0.943 | |
| TD | 3 | 0.948 | |

**Table 11.** *Cont.*

| Indicators | Number of Items | Cronbach's Alpha | Overall Cronbach's Alpha |
|:---:|:---:|:---:|:---:|
| TI | 3 | 0.950 | 0.972 |
| SS | 3 | 0.942 | |

**Table 12.** Test results.

| KMO and Bartlett's Test | | Results |
|:---:|:---:|:---:|
| KMO Sample Suitability Quantity | | 0.956 |
| | Approximate chi-square | 10,533.509 |
| Bartlett's test of sphericity | (Number of) degrees of freedom | 435 |
| | Significance | 0.000 |

### 4.3.2. Factor Analysis for Dimensionality Reduction

Previous studies have suggested that the DBSCAN algorithm of the two-stage clustering algorithm is most effective with low-dimensional data. Using this algorithm with high-dimensional data has become increasingly challenging. Therefore, in this study, we utilized factor analysis to downscale the diverse services before density clustering.

We employed principal component analysis to extract the common factors with eigenvalues larger than 1. The cumulative variance contribution of the common factors is 77.300%, indicating that the questionnaire has good structural validity. Factor analysis was conducted with the maximum variance method of rotation. The three common factors generated are shown in Table 13. The loading values of all indicators for each factor are greater than 0.5, and no indicators were eliminated.

**Table 13.** Rotated factor loading matrix.

| Survey Questions | Factor | | |
|:---|:---:|:---:|:---:|
| | 1 | 2 | 3 |
| TI1: The services provided by the park, such as science, technology and innovation, have a significant role to play. | 0.869 | | |
| SS3: The special services provided by the park are trustworthy and high quality. | 0.861 | | |
| TD1: The services provided by the park, such as talent recruitment, training and settlement, have a significant role to play. | 0.859 | | |
| SS1: The special services provided by the park are varied. | 0.855 | | |
| SS2: The special services provided by the park play a significant role in helping enterprises solve practical problems. | 0.851 | | |
| TI2: The science, technology and innovation services provided by the park are rich in content and highly professional. | 0.848 | | |
| TD2: The talent recruitment, training and settlement services provided by the park are highly professional and reliable. | 0.848 | | |
| TI3: The science, technology and innovation services provided by the park are timely and courteous. | 0.840 | | |
| SI2: The park has a wide variety of external publicity activities and channels. | 0.827 | | |
| FS1: The financing services provided by the park play a significant role. | 0.805 | | |
| TD3: The park responds quickly and solve problems in talent recruitment, training, settlement and other services. | 0.805 | | |
| FS2: The financing services provided by the park are highly specialized and reliable. | 0.794 | | |
| SI1: The brand of the park has a significant impact and can attract companies. | 0.782 | | |
| SI3: The service staff displays a professional and positive approach towards external promotional work. | 0.743 | | |

**Table 13.** *Cont.*

| Survey Questions | Factor | | |
|---|---|---|---|
| | 1 | 2 | 3 |
| FS3: The financing services provided by the park are responsive and can quickly help companies solve their problems. | 0.704 | | |
| BS2: The business service facilities and equipment provided by the park, including conference facilities and business travel for work-related trips, are user-friendly, efficient, and speedy. | | 0.883 | |
| BP1: The service staff for property warranty, parking payment, and other property services is courteous. | | 0.866 | |
| BP2: The property warranty, parking payment and other property services provided by the park are responsive and solve problems quickly. | | 0.866 | |
| BS1: The business service facilities and equipment provided by the park, including conference facilities and business travel for work-related trips, can meet a variety of needs. | | 0.865 | |
| IS2: The infrastructure facilities of the park are easy to use, simple and quick. | | 0.769 | |
| IS3: The infrastructure facilities of the park are safe, reliable and trustworthy. | | 0.726 | |
| BS3: Business services, including conference facilities and business travel, have a quality staff. | | 0.721 | |
| IS1: The infrastructure of the park is well-equipped to cater to various needs. | | 0.710 | |
| BP3: The park provides a credible and guaranteed commitment to property warranty and parking payment. | | 0.704 | |
| SM2: The park plays a significant role in responding to various emergencies and ensuring safety within the park. | | | 0.839 |
| SM3: The park is highly specialized and reliable in responding to various emergencies and ensuring safety within the park. | | | 0.816 |
| SM1: The park reacts quickly and disposes appropriately when dealing with various emergencies. | | | 0.804 |
| SG3: The degree of regularization and standardization of the service system in the park is increasing. | | | 0.770 |
| SG1: The internal system of the park is well organized and can provide stable services. | | | 0.756 |
| SG2: Enterprises can easily access the service specifications of the park and other relevant information. | | | 0.718 |

In Table 13, Factor 1 mainly includes services that are beneficial to the long-term development of the park and the enterprises, i.e., financing services, social influence, talent development, technology innovation, and specialty services. Therefore, Factor 1 is referred to as comprehensive development services. Factor 2 mainly includes services that provide basic protection for the enterprises, i.e., property services, infrastructure, and business meetings. Therefore, Factor 2 is referred to as basic protection services. Factor 3 mainly reflects the management level of the park itself and its ability to respond to emergencies. Therefore, Factor 3 is referred to as park management services. The three factors are as follows: the basic protection services dimension, park management services dimension, and comprehensive development services dimension.

4.3.3. Satisfaction Evaluation Based on Two-Stage Clustering

After dimensionality reduction, the clustering results for each park were analyzed in detail. Parks A and C were used as examples.

1.    A two-stage cluster analysis of CS about park A

There were a total of 106 valid questionnaires for park A. Throughout the clustering process, the distance calculation of the KNN algorithm was used to determine the inflection point of 1.03 at K = 4. With an input radius of $\varepsilon$ = 1.03 and a threshold of MinPts = 4, the clustering was more satisfactory, and the satisfaction results were clustered into one cluster. A total of seven noise points were generated during the process. Figure 4a,b show the satisfaction results for park A before and after eliminating the noise points in three dimensions. In Figure 4, the coordinate origin is (0, 0, 0), the X-axis represents the

basic protection services dimension, the Y-axis represents the park management services dimension, and the Z-axis represents the comprehensive development services dimension.

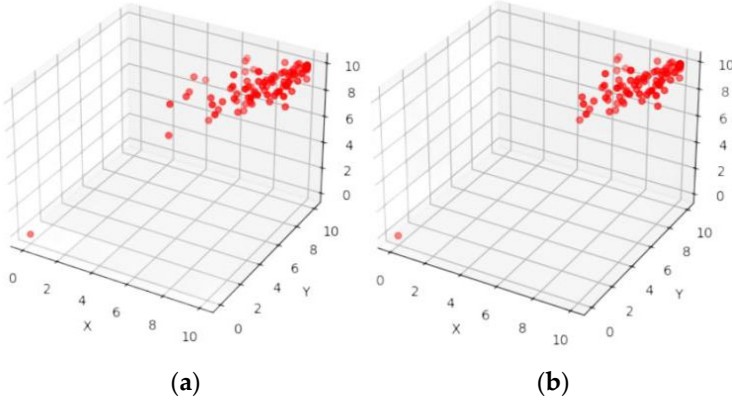

(**a**)                (**b**)

**Figure 4.** CSE results for park A (**a**) before eliminating outliers; (**b**) after eliminating outliers.

The above results show that the two-stage clustering algorithm eliminates the noise points, and the satisfaction clustering results of diverse services in park A are more reasonable. At present, the center point of clustering in park A is (8.92, 8.74, 9.27), indicating that the settled enterprises are highly satisfied with the services in all three dimensions.

From the dimension scores, it can be concluded that park A is not developing equally in the three dimensions. In comparison, park A has higher satisfaction in the comprehensive development dimension and lower satisfaction in the park management dimension.

2.    A two-stage cluster analysis of CS about park C

There were a total of 83 valid questionnaires for park C. Throughout the clustering process, the distance calculation of the KNN algorithm was used to determine the inflection point of 1.57 at K = 4. With an input radius of $\varepsilon$ = 1.57 and a threshold of MinPts = 4, the clustering was more satisfactory and the satisfaction results were clustered into one cluster. A total of five noise points were generated during the process. Figure 5a,b show the satisfaction results of park C before and after eliminating the noise points in three dimensions.

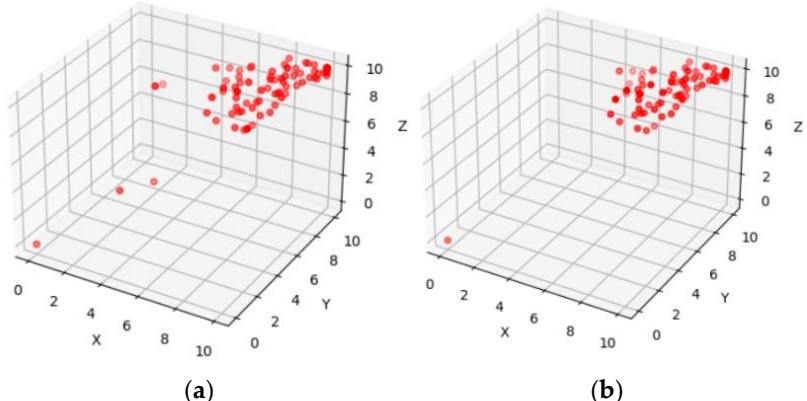

(**a**)                (**b**)

**Figure 5.** CSE results for park C (**a**) before eliminating outliers; (**b**) after eliminating outliers.

The above results show that the two-stage clustering algorithm eliminates the noise points, and the satisfaction clustering results of diverse services in this park are more reasonable. At present, the center point of clustering in park C is (8.12, 8.48, 9.11), indicating that the settled enterprises are highly satisfied with the services in all three dimensions. From the dimension scores, it can be concluded that park C is not developing equally in the three dimensions. In comparison, park C has higher satisfaction in the comprehensive development dimension and lower satisfaction in the basic protection dimension.

4.3.4. Analysis and Discussion

The results of the satisfaction ranking on the three dimensions before and after the clustering of each park are shown in Table 14. (1) In the dimension of basic protection services, park B > park A > park E > park C > park D > park F, (2) in the dimension of park management services, park B > park A > park C > park E > park D > park F, and (3) in the dimension of comprehensive development services, park A > park B > park C > park E > park D > park F.

**Table 14.** Satisfaction for each park before and after clustering in each dimension.

| Dimension | | Park A | Park B | Park C | Park D | Park E | Park F |
|---|---|---|---|---|---|---|---|
| Basic protection | Before clustering | 8.70 | 8.83 | 7.90 | 6.99 | 7.89 | 6.49 |
| | After clustering | 8.92 | 8.93 | 8.12 | 7.11 | 8.38 | 5.88 |
| Park management | Before clustering | 8.59 | 8.63 | 8.25 | 7.58 | 7.82 | 6.60 |
| | After clustering | 8.74 | 8.83 | 8.48 | 7.89 | 8.36 | 6.20 |
| Comprehensive development | Before clustering | 9.20 | 8.83 | 8.96 | 8.63 | 8.61 | 8.79 |
| | After clustering | 9.27 | 9.14 | 9.11 | 8.70 | 8.95 | 8.59 |

Based on Table 14, the analysis is as follows. (1) The ranking of the parks in each dimension changed before and after clustering. For example, before clustering, park C had a basic protection score of 7.90, which is higher than that of park E. After eliminating outliers, park C had a basic protection score of 8.12, which is lower than that of park E. (2) Satisfaction scores in the three dimensions were not equal for each park. Six parks had the highest scores in the comprehensive development services dimension. Parks A, B, and E had the lowest scores in the park management services dimension. Parks C, D, and F had the lowest scores in the basic protection services dimension.

In a practical sense, the results of the study indicate that the evaluation of park satisfaction is a multi-dimensional process. In this way, the park can improve its low-scoring services and achieve balanced development of its multiple services. Theoretically, the two-stage clustering algorithm corrects the satisfaction results of diverse services and solves the problem of "it is not reasonable for individual users to evaluate the satisfaction of diverse services", resulting in higher consensus after clustering.

*4.4. Comprehensive Evaluation of CS Based on SEM and Two-Stage Clustering in BTB Mode*

We compared the satisfaction results obtained from the two methods above. As shown in Table 15, in the overall satisfaction evaluation based on SEM, the ranking of satisfaction for each park is as follows: park A > park C > park B > park F > park D > park E. In the satisfaction evaluation of diverse services based on two-stage clustering, the ranking of satisfaction for each park is as follows: park A > park B > park C > park E > park D > park F. The satisfaction rankings obtained using the two methods are different.

**Table 15.** Comparison of satisfaction results between SEM and two-stage clustering.

| Methods | Park A | Park B | Park C | Park D | Park E | Park F |
|---|---|---|---|---|---|---|
| SEM | 9.25 | 8.88 | 8.91 | 8.76 | 8.34 | 8.77 |
| Two-stage clustering | 8.98 | 8.97 | 8.57 | 7.90 | 8.56 | 6.89 |

In this study, we invited seven experts comprising park managers, enterprise executives, and experts in the field of CS. The weights of the two evaluation methods in studying BTB CS were determined using the expert evaluation method. These two methods are satisfaction evaluation based on SEM and the satisfaction evaluation of diverse services based on two-stage clustering. Ultimately, we weighted the two methods to determine the final satisfaction score for each park. The formula is as follows:

$$CS \ of \ BTB = ETCS \times \omega_1 + TSC \times \omega_2 \tag{4}$$

where $CS \ of \ BTB$ denotes the final satisfaction score, $ETCS$ denotes the customer satisfaction index score for enterprise users based on SEM, and $\omega_1$ denotes the weight of the method. $TSC$ denotes the satisfaction score of diverse services based on two-stage clustering, and $\omega_2$ denotes the weight of the method.

In this case, the final satisfaction scores for each park are as follows: park A—9.086; park B—8.932; park C—8.706; park D—8.247; park E—8.474; and park F—7.642. The final ranking of each park's satisfaction score is as follows: park A > park B > park C > park E > park D > park F.

Combined with the field interviews, the evaluation results obtained using the comprehensive evaluation model are more appropriate for the actual situation and are beneficial to the rational allocation of service resources.

## 5. Conclusions

### 5.1. Findings

1.  Traditional CSE models are flawed in the BTB mode. In this study, we explored the applicability of traditional CSE models in the BTB mode. The theoretical differences between BTC and BTB in CSE were explored in terms of existing research, model validation, and case studies.
2.  The two-stage clustering algorithm can solve the problem of "it is not reasonable for individual users to evaluate the satisfaction of diverse services in the BTB mode". The clustering algorithm was mainly utilized to eliminate outlier data to achieve a more accurate satisfaction consensus. In this study, we optimized the individual user satisfaction scores by constructing a diverse service indicator system and using two-stage clustering combining the KNN algorithm and the DBSCAN algorithm. This step is concerned with the optimization of the evaluation results by the identification and elimination of noise points through the application of clustering algorithms. For example, a total of seven noise points were generated in park A during the optimization process, and the evaluation results are more reasonable after the elimination of the noise points.

### 5.2. Significance of the Study

1.  Theoretical significance. On the one hand, the shortcomings of the traditional BTC CSE models in the BTB mode have been verified in this article. On the other hand, a satisfaction evaluation model applicable to enterprise users has been proposed. The proposal and application of the new model provide the theoretical and applied foundations for subsequent research on CSE.
2.  Practical significance. Reasonable evaluation methods can improve the representativeness of evaluation results and the rationalization of resource allocation. This article can help researchers to recognize the shortcomings of traditional CSE models and to choose an appropriate evaluation model for conducting satisfaction evaluation studies. In addition, this study employed artificial intelligence technology, such as clustering algorithms, to evaluate the satisfaction in industrial parks. This ensured precise and dependable outcomes. Managers can rationally allocate resources and promote the sustainable development of settled enterprises through the use of artificial intelligence technology.

### 5.3. Future Prospects

We have solved the existing problems of CSE by applying a new algorithm. However, there are still some shortcomings that need to be addressed.

Firstly, the choice of clustering algorithm can be further optimized. We used a two-stage clustering algorithm to optimize the results of CSE. Since the algorithm is most effective on low-dimensional data, the original data were downscaled. The three dimensions after dimensionality reduction cover the main information of the original data. Subsequent studies may attempt to extend the dimensionality to four or more dimensions

for clustering. For example, a new method combined with improved DBSCAN and a density peak algorithm can be used to optimize the existing satisfaction evaluation results for industrial parks.

Secondly, the diverse service indicator system constructed in this study contains a total of 10 service components. Different settled enterprises may be more familiar with some services and less familiar with others. Therefore, the selection of respondents had an impact on our data.

Finally, the selection of cases for empirical studies could be further expanded. The research objects of this study were industrial parks and settled enterprises. They represent a common BTB mode where service providers offer diverse services to enterprise users. Satisfaction evaluation studies can be subsequently applied to other BTB modes.

**Author Contributions:** Conceptualization, methodology, investigation, writing—review and editing, supervision, J.W.; methodology, data curation, formal analysis, investigation, software, writing—original draft, writing—review and editing, L.Y. All authors have read and agreed to the published version of the manuscript.

**Funding:** This research was funded by Department of Social Sciences, Ministry of Education, grant number 22YJA630082. The program is Research Project of Humanities and Social Sciences of the Ministry of Education, China.

**Institutional Review Board Statement:** Not applicable.

**Informed Consent Statement:** Not applicable.

**Data Availability Statement:** Available by request from the corresponding author.

**Conflicts of Interest:** The authors declare no conflicts of interest.

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
