# Peer review of "Optimizing Business-to-Business Customer Satisfaction Analysis through Advanced Two-Stage Clustering: Insights from Industrial Parks"

_sustainability, doi:10.3390/su16104043_

Round 1

Reviewer 1 Report

Comments and Suggestions for Authors

Research on Customer Satisfaction Evaluation in BTB with Diverse Services Based on Two-Stage Clustering Algorithm Optimization: A Case of Industrial Parks

sustainability-2948120-peer-review-v1

Thank you for the opportunity to read this paper. This paper represents a valuable contribution to understanding and evaluating customer satisfaction in BTB contexts, especially within industrial parks. Addressing the suggested improvements could enhance its impact and clarity, ensuring its relevance to both academic and practitioner audiences. Before recommending this paper for publication I have the following comments:

Title

Given the innovative approach of using a two-stage clustering algorithm for evaluating customer satisfaction in a BTB context with diverse services, particularly in the setting of industrial parks, a title that succinctly captures the essence of this novel method while maintaining clarity on the study's scope and context could enhance its appeal and specificity. Here’s a suggestion:

"Optimizing BTB Customer Satisfaction Analysis through Advanced Two-Stage Clustering: Insights from Industrial Parks"

Abstract

To further align with the body of the paper, the abstract could include more specific insights or quantitative results from the study, which would provide a snapshot of the study's impact and findings.

1. Introduction

To strengthen alignment, it could more explicitly tie the research questions and objectives back to the stated need for innovation in methodology, as outlined in the title and abstract.

  • Specificity and Objectives: While the introduction outlines the research context, specifying the objectives in a numbered or bulleted list could enhance clarity and focus.
  • Research Questions: Explicitly stating research questions can guide the reader through the investigation's aim.

2. Literature Review

To enhance coherence with the study's innovative approach, it may benefit from a deeper exploration of previous methodologies' limitations in BTB contexts and a clearer linkage to how the proposed method addresses these gaps.

  • Critique of Existing Models: Expand on the limitations of existing models by including comparative analysis or case studies where these models may have fallen short in BTB settings.
  • Synthesis: Strengthen the literature review by synthesizing findings to directly build towards the necessity of the proposed two-stage clustering algorithm.

3. Methodology

For better alignment, it could further clarify how this approach directly responds to the challenges identified in the literature review, specifically in terms of improving upon traditional CSE methods in BTB settings.

  • Methodological Justification: Elaborate on the choice of the two-stage clustering algorithm over other potential methods. Why are KNN and DBSCAN specifically suited for this study’s aims?
  • Sample Selection: Provide more details on the selection of industrial parks and the criteria for choosing respondents within these parks.

4. Results

Including comparative analysis or direct references to how these results overcome limitations of traditional methods could enhance the narrative flow and coherence across sections.

  • Visualization: Incorporate more graphical representations of data, especially for clustering results, to enhance interpretability.
  • Comparative Analysis: Include a more detailed comparison between results obtained from traditional methods and the proposed model, highlighting specific improvements or insights gained.

5. Discussion

It could be improved by more directly referencing the initial research questions or objectives stated in the introduction, ensuring a full-circle narrative.

  • Link to Theory: Strengthen connections between empirical findings and theoretical implications. How do results inform our understanding of customer satisfaction dynamics in BTB settings?
  • Practical Implications: Elaborate on actionable insights for managers of industrial parks and other BTB service providers.

6. Conclusions and Future Work

Ensuring that the conclusions directly reflect the innovative approach highlighted in the title and abstract can enhance the paper's overall coherence.

  • Limitations: Discuss more comprehensively the limitations of the study, including potential biases in respondent selection and the generalizability of findings.
  • Future Research Directions: Besides exploring other algorithms and contexts, suggest specific methodologies or data sources that could further validate or challenge the study’s conclusions.

In conclusion, highlighting the innovative aspect of the methodology in each section, in line with the emphasis on innovation in the title, will strengthen the alignment and impact of the paper.

Refrences

  • Integrating Recent References: For instance, if a 2023 reference discusses the growing complexity of customer needs in BTB relationships, use it to underscore the necessity for your advanced clustering approach in capturing these nuanced satisfaction metrics.
  • Addressing Gaps: If recent literature (e.g., 2022 or 2023 references) highlights gaps in the application of traditional customer satisfaction models in BTB contexts, explicitly connect these observations to how your research methodology addresses these gaps.

Comments on the Quality of English Language

Reviewer 2 Report

Comments and Suggestions for Authors

This is an interesting manuscript. The research ideas of the paper are relatively clear, the selection of research methods is reasonable, and it has a certain degree of innovation. However, there are still the following shortcomings in the current paper:

1. The literature review in the introduction is insufficient, and there is a lack of research context and shortcomings.

2. The results discussion and analysis section should further clarify the comparative analysis with current research results of the same type, highlight the new results and research value of this article, and broaden the research perspective.

3. In the summary section of the article, further clarification should be provided on the relevant theories proposed in the paper

Author Response

请参阅附件。

Reviewer 3 Report

Comments and Suggestions for Authors

The article is on an exciting and up-to-date topic. There are not many publications dealing with the covered issues. The text is full of tables and graphs and "real research, " which is a big plus for the article. However, the article has several weaknesses. The author should summarize the existing state of knowledge in a given area and cite the sources. In the introduction, it would be OK to add the used methodology and define how these methods were used to achieve the aim of the article.

On the other hand, it's pretty clear from the text. Formally, the keywords are not selected appropriately. But it's all just minimal objections that can not downgrade the level of the article.
The overall merit of the article is very high, and its research is beneficial for expert readers. It fills up the gap in the existing research considering the presented field of the article, and thus, I can recommend this article to be published in its current form.

Author Response

请参阅附件。

Reviewer 4 Report

Comments and Suggestions for Authors

Dear authors! The article is certainly relevant. The article will greatly benefit if the authors clearly indicate the purpose of the research, and also present the scientific and practical implications of the presented research. It is important to show how the presented methods can ensure representativeness of the analytics. In conclusion, it is important to show the theoretical conclusions and the obtained theoretical and practical provisions in the format of scientific novelty.

Author Response

请参阅附件。

Round 2

Reviewer 1 Report

Comments and Suggestions for Authors

Dear Authors,

I want to commence by expressing my appreciation for the authors' diligent efforts in revising their manuscript. The revisions undertaken reflect a commendable commitment to enhancing the quality and clarity of this research, demonstrating a proactive approach to addressing the feedback provided. The study's subject matter is of significant relevance, and the enhancements made thus far underscore the potential impact of this work within its field. However, to recommend the manuscript for publication in this journal, higher standards of academic writing are required. I strongly recommend that the authors seek professional English editing services (see sample observations attached). Ensuring the manuscript undergoes a thorough review by a professional, especially one with expertise in academic writing within this specific domain, will be invaluable. Proof of the editing certificate is also requested upon completion of this process. This step will not only refine the readability and coherence of the manuscript but also reinforce its contribution to the scholarly community.

Comment 3:

The revised paper clearly outlines the research questions: "(1) Does the traditional BTC CSE model apply to the BTB mode in the context of diverse services? (2) What methods are used to optimize the results of individual user satisfaction scoring in the BTB mode, considering the difference between group scoring and individual scoring?”. This directly responds to the initial comment suggesting the need for explicitly stated research questions to guide the reader through the investigation's aim.

Regarding the specificity and objectives, the revision provides a clearer understanding of the research's aim and the problems it intends to address by developing a new model and introducing a novel algorithm for a CSE model applicable to the BTB mode. This approach enhances clarity and focus, aligning with the suggestion to specify objectives in a numbered or bulleted list for improved clarity and focus.

Overall, the revisions made in the document effectively address the comments related to the specificity of objectives and the explicit statement of research questions, enhancing the paper's clarity and focus on guiding the reader through the investigation's aims.

Comment 5:

The revised version of the paper addresses the comment regarding methodological justification for the choice of the two-stage clustering algorithm over other potential methods, and specifically, why KNN and DBSCAN are suited for the study's aims. The paper explains the choice of a two-stage clustering algorithm that combines KNN (K-Nearest Neighbor) classification algorithm and DBSCAN (Density-Based Spatial Clustering of Applications with Noise) to optimize the BTB Customer Satisfaction Evaluation (CSE).

DBSCAN: It is described as a classical density-based clustering algorithm advantageous because it doesn't require predetermining the number of clusters. It's effective in identifying arbitrarily shaped data sets and outliers. This capability is essential for the study as it aims to identify and eliminate noise point data from individual user satisfaction evaluation results, leading to higher consensus in the final clustering results.

KNN: The K-Nearest Neighbor algorithm is used to determine the classification of a sample by calculating its distance to all samples, thereby reducing the iterative experimental process of input parameters in the DBSCAN clustering algorithm. This reduction is achieved by confirming the input radius, which decreases human interference in the process. The use of the KNN algorithm in conjunction with DBSCAN helps optimize the latter by providing a more efficient way to set the input parameters, making it a suitable choice for the study's objectives. 

The paper elaborates on how these algorithms are integrated into a two-stage clustering process to optimize satisfaction evaluation results, significantly contributing to the methodological justification for their selection. This detailed explanation directly addresses the comment on elaborating the choice of these specific algorithms, highlighting their suitability for achieving the study's aims related to optimizing BTB CSE.

Comments 8:

The revised version of the paper addresses the comment on linking to theory and outlining practical implications by elaborating on the significance of the study in both theoretical and practical domains:

Theoretical significance: The paper acknowledges the gaps in traditional BTC (Business to Customer) Customer Satisfaction Evaluation (CSE) models when applied to the BTB (Business to Business) mode and introduces a new satisfaction evaluation model suitable for enterprise users. This innovation provides a foundational framework for future research in CSE, particularly in contexts where traditional models fall short.

Practical significance: The study emphasizes how the application of an appropriate evaluation model, coupled with the use of advanced algorithms like clustering, can enhance the accuracy and representativeness of satisfaction evaluation results. This approach allows for a more rational allocation of resources and promotes sustainable development among settled enterprises, leveraging artificial intelligence technology for evaluating satisfaction in industrial parks.

These sections in the paper directly respond to the request for a stronger link to underlying theoretical principles and a clear articulation of the study's real-world implications, demonstrating that the revised paper has addressed these comments effectively.

Thank you!

Comments on the Quality of English Language

See attached PDF file

Author Response

请参阅附件。

Round 3

Reviewer 1 Report

Comments and Suggestions for Authors

The authors have replied to my concerns satisfactorily.